# Life-Long Disentangled Representation Learning with Cross-Domain Latent Homologies

**Alessandro Achille, Tom Eccles, Loic Matthey, Christopher P Burgess,**
**Nick Watters, Alexander Lerchner, Irina Higgins**
UCLA, DeepMind
achille@cs.ucla.edu,
{eccles,lmatthey,cpburgess,nwatters,lerchner,irinah}@google.com

## Abstract

Intelligent behaviour in the real-world requires the ability to acquire new knowledge from an ongoing sequence of experiences while preserving and reusing past knowledge. We propose a novel algorithm for unsupervised representation learning from piece-wise stationary visual data: Variational Autoencoder with Shared Embeddings (VASE). Based on the Minimum Description Length principle, VASE automatically detects shifts in the data distribution and allocates spare representational capacity to new knowledge, while simultaneously protecting previously learnt representations from catastrophic forgetting. Our approach encourages the learnt representations to be disentangled, which imparts a number of desirable properties: VASE can deal sensibly with ambiguous inputs, it can enhance its own representations through imagination-based exploration, and most importantly, it exhibits semantically meaningful sharing of latents between different datasets. Compared to baselines with entangled representations, our approach is able to reason beyond surface-level statistics and perform semantically meaningful cross-domain inference.

## 1 Introduction

A critical feature of biological intelligence is its capacity for *life-long learning* [10] – the ability to acquire new knowledge from a sequence of experiences to solve progressively more tasks, while maintaining performance on previous ones. This, however, remains a serious challenge for current deep learning approaches. While current methods are able to outperform humans on many individual problems [53, 37, 20], these algorithms suffer from *catastrophic forgetting* [14, 34, 35, 43, 17]. Training on a new task or environment can be enough to degrade their performance from super-human to chance level [47]. Another critical aspect of life-long learning is the ability to sensibly reuse previously learnt representations in new domains (*positive transfer*). For example, knowing that strawberries and bananas are not edible when they are green could be useful when deciding whether to eat a green peach in the future. Finding semantic homologies between visually distinctive domains can remove the need to learn from scratch on every new environment and hence help with data efficiency – another major drawback of current deep learning approaches [16, 30].

But how can an algorithm maximise the informativeness of the representation it learns on one domain for positive transfer on other domains without knowing a priori what experiences are to come? One approach might be to capture the important structure of the current environment in a maximally compact way (to preserve capacity for future learning). Such learning is likely to result in positive transfer if future training domains share some structural similarity with the old ones. This is a reasonable expectation to have for most natural (non-adversarial) tasks and environments, since they tend to adhere to the structure of the real world (e.g. relate to objects and their properties) governed by the consistent rules of chemistry or physics. A similar motivation underlies the Minimum Description Length (MDL) principle [45] and disentangled representation learning [8].

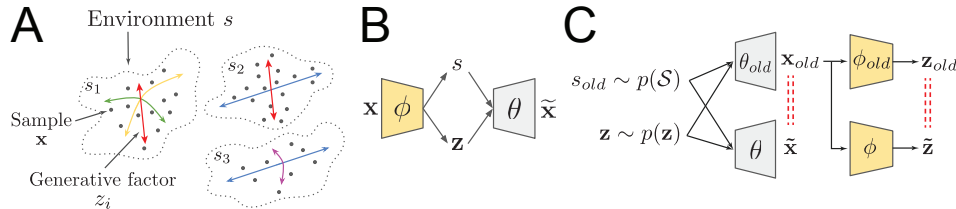

Figure 1: **A**: Schematic representation of the life-long learning data distribution. Each dataset/environment corresponds to a cluster $s$. Data samples $\mathbf{x}$ constituting each cluster can be described by a local set of coordinates (data generative factors $z_n$). Different clusters may share some data generative factors. **B**: VASE model architecture **C**: Schematic of the "dreaming" feedback loop. We use a snapshot of the model with the old parameters ($\phi_{old}$, $\theta_{old}$) to generate an imaginary batch of data $\mathbf{x}_{old}$ for a previously experienced dataset $s_{old}$. While learning in the current environment, we ensure that the representation is still consistent on the hallucinated "dream" data, and can reconstruct it (see red dashed lines).

Recent state of the art approaches to unsupervised disentangled representation learning [21, 9, 25, 29] use a modified Variational AutoEncoder (VAE) [27, 44] framework to learn a representation of the data generative factors. These approaches, however, only work on independent and identically distributed (IID) data from a single visual domain. This paper extends this line of work to life-long learning from piece-wise stationary data, exploiting this setting to learn shared representations across domains where applicable. The proposed Variational Autoencoder with Shared Embeddings (VASE, see fig. 1B) automatically detects shifts in the training data distribution and uses this information to allocate spare latent capacity to novel dataset-specific disentangled representations, while reusing previously acquired representations of latent dimensions where applicable. We use latent masking and a generative "dreaming" feedback loop (similar to [42, 51, 50, 5]) to avoid catastrophic forgetting. Our approach outperforms [42], the only other VAE based approach to life-long learning we are aware of. Furthermore, we demonstrate that the pressure to disentangle endows VASE with a number of useful properties: 1) dealing sensibly with ambiguous inputs; 2) learning richer representations through imagination-based exploration; 3) performing semantically meaningful cross-domain inference by ignoring irrelevant aspects of surface-level appearance.

## 2 Related work

The existing approaches to continual learning can be broadly separated into three categories: data-, architecture- or weights-based. The data-based approaches augment the training data on a new task with the data collected from the previous tasks, allowing for simultaneous multi-task learning on IID data [11, 46, 43, 34, 15]. The architecture-based approaches dynamically augment the network with new task-specific modules, which often share intermediate representations to encourage positive transfer [47, 40, 48, 49]. Both of these types of approaches, however, are inefficient in terms of the memory requirements once the number of tasks becomes large. The weights-based approaches do not require data or model augmentation. Instead, they prevent catastrophic forgetting by slowing down learning in the weights that are deemed to be important for the previously learnt tasks [28, 55, 39]. This is a promising direction, however, its application is limited by the fact that it typically uses knowledge of the task presentation schedule to update the loss function after each switch in the data distribution.

Most of the continual learning literature, including all of the approaches discussed above, have been developed in task-based settings, where representations are learnt implicitly. While deep networks learn well in such settings [1, 52], this often comes at a cost of reduced positive transfer. This is because the implicitly learnt representations often overfit to the training task by discarding information that is irrelevant to the current task but may be required for solving future tasks [1, 2, 3, 52, 22]. The acquisition of useful representations of complex high-dimensional data without task-based overfitting is a core goal of unsupervised learning. Past work [2, 4, 21] has demonstrated the usefulness of information-theoretic methods in such settings. These approaches can broadly be seen as efficient implementations of the Minimum Description Length (MDL) principle for unsupervised learning [45, 18]. The representations learnt through such methods have been shown to help in transfer scenarios and with data efficiency for policy learning in the Reinforcement Learning (RL) context [22]. These approaches, however, do not immediately generalise to non-stationary data. Indeed, life-long unsupervised representation learning is relatively under-developed [51, 50, 39]. The majority of recent work in this direction has concentrated on implicit generative models [51, 50], or non-parametric

approaches [36]. Since these approaches do not possess an inference mechanism, they are unlikely to be useful for subsequent task or policy learning. Furthermore, none of the existing approaches explicitly investigate meaningful sharing of latent representations between environments.

## 3  Framework

### 3.1  Problem formalisation

We assume that there is an a priori unknown set $\mathcal{S} = \{s_1, s_2, ..., s_K\}$ of $K$ environments which, between them, share a set $\mathcal{Z} = \{z_1, z_2, ..., z_N\}$ of $N$ independent data generative factors. We assume $\mathbf{z} \sim \mathcal{N}(\mathbf{0}, \mathbf{I})$. Since we aim to model piece-wise stationary data, it is reasonable to assume $s \sim \mathrm{Cat}(\pi_{1,...,K})$, where $\pi_k$ is the probability of observing environment $s_k$. Two environments may use the same generative factors but render them differently, or they may use a different subset of factors altogether. Given an environment $s$, and an environment-dependent subset $\mathcal{Z}^s \subseteq \mathcal{Z}$ of the ground truth generative factors, it is possible to synthesise a dataset of images $\mathbf{x}^s \sim p(\cdot | \mathbf{z}^s, s)$. In order to keep track of which subset of the $N$ data generative factors is used by each environment $s$ to generate images $\mathbf{x}^s$, we introduce an environment-dependent mask $\mathbf{a}^s$ with dimensionality $|\mathbf{a}| = N$, where $a_n^s = 1$ if $z_n \in \mathcal{Z}^s$ and zero otherwise. A similar masking has also been used by [24] to enforce disentanglement in a single environment, but assuming additional side-information about the generative factors. Hence, we assume $\mathbf{a}^s \sim \mathrm{Bern}(\omega_{1,...,N}^s)$, where $\omega_n^s$ is the probability that factor $z_n$ is used in environment $s$. This leads to the following generative process (where "$\odot$" is element-wise multiplication):

$$\mathbf{z} \sim \mathcal{N}(\mathbf{0}, \mathbf{I}), \qquad s \sim \mathrm{Cat}(\pi_{1,...,K}), \qquad \mathbf{a}^s \sim \mathrm{Bern}(\omega_{1,...,N}^s),$$
$$\mathbf{z}^s = \mathbf{a}^s \odot \mathbf{z}, \qquad \mathbf{x}^s \sim p(\cdot \,|\, \mathbf{z}^s, s) \tag{1}$$

Intuitively, we assume that the piece-wise stationary observed data $\mathbf{x}$ can be split into *clusters* (environments $s$) (note evidence for similar experience clustering from the animal literature [6]). Each cluster has a set of *standard coordinate axes* (a subset of the generative factors $\mathbf{z}$ chosen by the latent mask $\mathbf{a}^s$) that can be used to parametrise the data in that cluster (fig. 1A). Given a sequence $\mathbf{x} = (\mathbf{x}^{s_1}, \mathbf{x}^{s_2}, ...)$ of datasets generated according to the process in eq. (1), where $s_k \sim p(\mathbf{s})$ is the $k$-th sample of the environment, the aim of life-long representation learning can be seen as estimating the full set of generative factors $\mathcal{Z} \approx \bigcup_k q(\mathbf{z}^{s_k} | \mathbf{x}^{s_k})$ from the environment-specific subsets of $\mathbf{z}$ inferred on each stationary data cluster $\mathbf{x}^{s_k}$. Henceforth, we will drop the subscript $k$ for simplicity of notation.

### 3.2  Inferring the data generative factors

Observations $\mathbf{x}^s$ cannot contain information about the generative factors $z_n$ that are not relevant for the environment $s$. Hence, we use the following form for representing the data generative factors:

$$q(\mathbf{z}^s | \mathbf{x}^s) = \mathbf{a}^s \odot \mathcal{N}(\mu(\mathbf{x}), \sigma(\mathbf{x})) + (1 - \mathbf{a}^s) \odot \mathcal{N}(\mathbf{0}, \mathbf{I}). \tag{2}$$

Note that $\mu$ and $\sigma$ in eq. (2) depend only on the data $\mathbf{x}$ and not on the environment $s$. This is important to ensure that the semantic meaning of each latent dimension $z_n$ remains consistent for different environments $s$. We model the representation $q(\mathbf{z}^s | \mathbf{x}^s)$ of the data generative factors as a product of independent normal distributions to match the assumed prior $p(\mathbf{z}) \sim \mathcal{N}(\mathbf{0}, \mathbf{I})$.

In order to encourage the representation $q(\mathbf{z}^s | \mathbf{x}^s)$ to be semantically meaningful, we encourage it to capture the generative factors of variation within the data $\mathbf{x}^s$ by following the MDL principle. We aim to find a representation $\mathbf{z}^s$ that minimises the reconstruction error of the input data $\mathbf{x}^s$ conditioned on $\mathbf{z}^s$ under a constraint on the quantity of information in $\mathbf{z}^s$. This leads to the following loss function:

$$\mathcal{L}_{\mathrm{MDL}}(\phi, \theta) = \underbrace{\mathbb{E}_{\mathbf{z}^s \sim q_\phi(\cdot | \mathbf{x}^s)}[-\log p_\theta(\mathbf{x} \,|\, \mathbf{z}^s, s)]}_{\text{Reconstruction error}} + \gamma \,|\underbrace{\mathbb{KL}(q_\phi(\mathbf{z}^s | \mathbf{x}^s) || p(\mathbf{z}))}_{\text{Representation capacity}} - \underbrace{C}_{\text{Target}}|^2 \tag{3}$$

The loss in eq. (3) is closely related to the $\beta$-VAE [21] objective $\mathcal{L} = \mathbb{E}_{\mathbf{z} \sim q_\phi(\cdot | \mathbf{x})}[-\log p_\theta(\mathbf{x} | \mathbf{z})] + \beta \,\mathbb{KL}(q_\phi(\mathbf{z} | \mathbf{x}) || p(\mathbf{z}))$, which uses a Lagrangian to limit the latent bottleneck capacity, rather than an explicit target $C$. It was shown that optimising the $\beta$-VAE objective helps with learning a more semantically meaningful disentangled representation $q(\mathbf{z} | \mathbf{x})$ of the data generative factors [21]. However, [9] showed that progressively increasing the target capacity $C$ in eq. (3) throughout training further improves the disentanglement results reported in [21], while simultaneously producing sharper

reconstructions. Progressive increase of the representational capacity also seems intuitively better suited to continual learning where new information is introduced in a sequential manner. Hence, VASE optimises the objective function in eq. (3) over a sequence of datasets $\mathbf{x}^s$. This, however, requires a way to infer $s$ and $\mathbf{a}^s$, as discussed next.

### 3.3    Inferring the latent mask

Given a dataset $\mathbf{x}^s$, we want to infer which latent dimensions $z_n$ were used in its generative process (see eq. (1)). This serves multiple purposes: 1) helps identify the environment $s$ (see next section); 2) helps ignore latent factors $z_n$ that encode useful information in some environment but are not used in the current environment $s$, in order to prevent retraining and subsequent catastrophic forgetting; and 3) promotes latent sharing between environments. Remember that eq. (3) indirectly optimises for $\mathbb{E}_{\mathbf{x}^s}[q_\phi(\mathbf{z}^s|\mathbf{x}^s)] \approx p(\mathbf{z})$ after training on a dataset $s$. If a new dataset uses the same generative factors as $\mathbf{x}^s$, then the marginal behaviour of the corresponding latent dimensions $z_n$ will not change. On the other hand, if a latent dimension encodes a data generative factor that is irrelevant to the new dataset, then it will start behaving atypically and stray away from the prior. We capture this intuition by defining the *atypicality* score $\alpha_n$ for each latent dimension $z_n$ on a batch of data $\mathbf{x}^s_{\text{batch}}$:

$$\alpha_n = \mathbb{KL}\big(\mathbb{E}_{\mathbf{x}^s_{\text{batch}}}[\, q_\phi(z_n^s|\mathbf{x}^s_{\text{batch}})\,] \,||\, p(z_n)\big). \tag{4}$$

The atypical components are unlikely to be relevant to the current environment, so we mask them out:

$$a_n^s = \left\{ \begin{array}{ll} 1, \text{if } \alpha_n < \lambda \\ 0, \text{otherwise} \end{array} \right. \tag{5}$$

where $\lambda$ is a threshold hyperparameter (see appendices A.2 and A.3 for more details). Note that the uninformative latent dimensions $z_n$ that have not yet learnt to represent any data generative factors, i.e. $q_\phi(z_n|\mathbf{x}_n^s) = p(z_n)$, are automatically unmasked in this setup (as they will have $\alpha_n \approx 0$). This allows them to be available as spare latent capacity to learn new generative factors when exposed to a new dataset. Fig. 2 (bottom third panel) shows the sharp changes in $\alpha_n$ at dataset boundaries during training.

### 3.4    Inferring the environment

Given the generative process introduced in eq. (1), it may be tempting to treat the environment $s$ as a discrete latent variable and learn it through amortised variational inference. However, we found that in the continual learning scenario this is not a viable strategy. Parametric learning is slow, yet we have to infer each new data cluster $s$ extremely fast to avoid catastrophic forgetting. Hence, we opt for a fast non-parametric meta-algorithm motivated by the following intuition. Having already experienced $r$ datasets during life-long learning, there are two choices when it comes to inferring the current one $s$: it is either a new dataset $s_{r+1}$, or it is one of the $r$ datasets encountered in the past. Intuitively, one way to check for the former is to see whether the current data $\mathbf{x}^s$ seems likely under any of the previously seen environments. This condition on its own is not sufficient though. First, it is possible that environment $s$ uses a subset of the generative factors used by another environment $\mathcal{Z}^s \subseteq \mathcal{Z}^t$, in which case environment $t$ will explain the data $\mathbf{x}^s$ well, yet it will be an incorrect inference. Hence, we have to ensure that the subset of the relevant generative factors $\mathbf{z}^s$ inferred for the current data $\mathbf{x}^s$ according to section 3.3 matches that of the candidate past dataset $t$. Second, finding a past dataset that matches the current one on the subset of the relevant generative factors without checking the reconstruction accuracy is not sufficient. For example, an environment with a moving square should not be classified as being the same as the environment with a moving triangle, despite the two environments sharing the same generative factors (the object position). Hence the reconstruction error should be involved in the inference.

Given the considerations above, we infer the environment $s$ for a batch $\mathbf{x}^s_{\text{batch}}$ according to:

$$s = \left\{ \begin{array}{ll} \hat{s} & , \text{ if } \mathbb{E}_{\mathbf{z}^{\hat{s}}}[\, p_\theta(\mathbf{x}^s_{\text{batch}}|\mathbf{z}^{\hat{s}},\hat{s})\,] \leq \kappa L_{\hat{s}} \,\wedge\, \mathbf{a}^s = \mathbf{a}^{\hat{s}} \\ s_{r+1}, & \text{otherwise} \end{array} \right. \tag{6}$$

where $\hat{s} = \arg\max_s q(s|\mathbf{x}^s_{\text{batch}})$ is the output of an auxiliary classifier trained to infer the most likely previously experienced environment $\hat{s}$ given the current batch $\mathbf{x}^s_{\text{batch}}$, $L_{\hat{s}}$ is the average reconstruction error observed for the environment $\hat{s}$ when it was last experienced, and $\kappa$ is a threshold hyperparameter (see appendix A.2 for details).

## 3.5 Preventing catastrophic forgetting

So far we have discussed how VASE integrates knowledge from the current environment into its representation $q_\phi(\mathbf{z}|\mathbf{x})$, but we haven't yet discussed how we ensure that past knowledge is not forgotten in the process. Most standard approaches to preventing catastrophic forgetting discussed in section 2 are either not applicable to a variational context, or do not scale well due to memory requirements. However, thanks to learning a generative model of the observed environments, we can prevent catastrophic forgetting by periodically *hallucinating* (i.e. generating samples) from past environments using a snapshot of VASE, and making sure that the current version of VASE is still able to model these samples. A similar "dreaming" feedback loop was used in [42, 51, 50, 5].

More formally, we follow the generative process in eq. (1) to create a batch of samples $\mathbf{x}_{\text{old}} \sim q_{\theta_{\text{old}}}(\cdot|\mathbf{z}, s_{\text{old}})$ using a snapshot of VASE with parameters $(\phi_{\text{old}}, \theta_{\text{old}})$ (see fig. 1C). We then update the current version of VASE according to the following (replacing $_{\text{old}}$ with $'$ for brevity):

$$\mathcal{L}_{\text{past}}(\phi,\theta) = \mathbb{E}_{\mathbf{z},s',\mathbf{x}'}\Big[ \underbrace{D[q_\phi(\mathbf{z}|\mathbf{x}'), q_{\phi'}(\mathbf{z}'|\mathbf{x}')]}_{\text{Encoder proximity}} + \underbrace{D[q_\theta(\mathbf{x}|\mathbf{z},s'), q_{\theta'}(\mathbf{x}'|\mathbf{z},s')]}_{\text{Decoder proximity}} \Big], \tag{7}$$

where $D$ is a distance between two distributions. For the decoder, which is a product of Bernoulli random variables, we use the KL divergence as the distance $D$. For the Gaussian encoder, we tried both the KL divergence and the Wasserstein distance $W = \|\mu_0 - \mu_1\|^2 + \|\Sigma_0^{1/2} - \Sigma_1^{1/2}\|^2$. We did not observe significant differences between the two distance metrics for the majority of the hyperparameter settings. However, we found the gradients of the Wasserstein distance $W$ to be better behaved. Hence, we use the Wasserstein distance for the encoder and the KL distance for the decoder in all experiments. The snapshot parameters get synced to the current trainable parameters $\phi_{\text{old}} \leftarrow \phi$, $\theta_{\text{old}} \leftarrow \theta$ every $\tau$ training steps, where $\tau$ is a hyperparameter. The expectation over simulators $s_{\text{old}}$ and latents $\mathbf{z}$ in eq. (7) is done using Monte Carlo sampling (see appendix A.2 for details).

## 3.6 Model summary

To summarise, we train our model using a meta-algorithm with both parametric and non-parametric components. The latter is needed to quickly associate new experiences to an appropriate cluster, so that learning can happen inside the current experience cluster, without disrupting unrelated clusters. We initialise the latent representation $\mathbf{z}$ to have at least as many dimensions as the total number of the data generative factors $|\mathbf{z}| \geq |\mathcal{Z}| = N$, and the softmax layer of the auxiliary environment classifier to be at least as large as the number of datasets $|\mathcal{S}| = K$. As we observe the sequence of training data, we detect changes in the environment and dynamically update the internal estimate of $r \leq K$ datasets experienced so far according to eq. (6). We then train VASE by minimising the following objective function:

$$\mathcal{L}(\phi,\theta) = \underbrace{\mathbb{E}_{\mathbf{z}^s \sim q_\phi(\cdot|\mathbf{x}^s)}[-\log p_\theta(\mathbf{x}|\mathbf{z}^s,s)] + \gamma|\mathbb{KL}(q_\phi(\mathbf{z}^s|\mathbf{x}^s)||p(\mathbf{z})) - C|^2}_{\text{MDL on current data}} +$$

$$+ \underbrace{\mathbb{E}_{\mathbf{z},s',\mathbf{x}'}\Big[ D[q_\phi(\mathbf{z}|\mathbf{x}'), q_{\phi'}(\mathbf{z}'|\mathbf{x}')] + D[q_\theta(\mathbf{x}|\mathbf{z},s'), q_{\theta'}(\mathbf{x}'|\mathbf{z},s')] \Big]}_{\text{"Dreaming" feedback on past data}}. \tag{8}$$

## 4 Experiments

**Continual learning with disentangled shared latents** First, we qualitatively assess whether VASE is able to learn good representations in a continual learning setup. We use a sequence of three datasets: (1) a moving version of Fashion-MNIST [54] (shortened to moving Fashion), (2) MNIST [31], and (3) a moving version of MNIST (moving MNIST). During training we expect VASE to detect shifts in the data distribution and dynamically create new experience clusters $s$, learn a disentangled representation of each environment without forgetting past environments, and share disentangled factors between environments in a semantically meaningful way. Fig. 2 (top) compares the performance of VASE to that of Controlled Capacity Increase-VAE (CCI-VAE) [9], a model for disentangled representation learning with the same architecture as VASE but without the modifications introduced in this paper to allow for continual learning. It can be seen that unlike VASE, CCI-VAE forgot moving Fashion at the end of the training sequence. Both models were able to disentangle position from object identity, however, only VASE was able to meaningfully share latents between the different datasets - the two positional latents

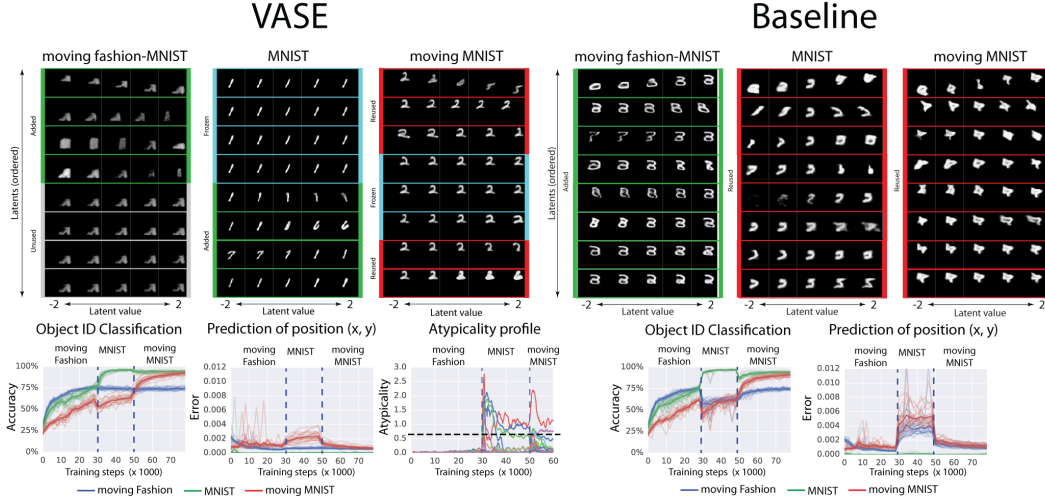

Figure 2: We compare VASE to a CCI-VAE baseline. Both are trained on a sequence of three datasets: moving fashion MNIST (moving Fashion) → MNIST → moving MNIST. **Top**: latent traversals at the end of training seeded with samples from the three datasets. The value of each latent $z_n$ is traversed between -2 and 2 one at a time, and the corresponding reconstructions are shown. Rows correspond to latent dimensions $z_n$, columns correspond to the traversal values. Latent use progression throughout training is demonstrated in colour. **Bottom**: performance of MNIST and Fashion object classifiers and a position regressor trained on the latent space **z** throughout training. Note the relative stability of the curves for VASE compared to the baseline. The atypicality profile shows the values of $\alpha_n$ through training (different colours indicate different latent dimensions), with the threshold $\lambda$ indicated by the dashed black line.

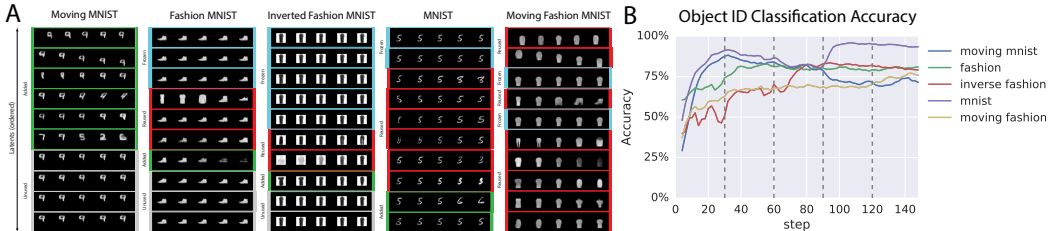

Figure 3: Latent traversals (**A**) and classification accuracy (**B**) (both as in fig. 2) for VASE trained on a sequence of moving MNIST → Fashion → inverse Fashion → MNIST → moving Fashion. See fig. 8 for larger traversals.

are active for two moving datasets but not for the static MNIST. VASE also has moving Fashion- and MNIST-specific latents, while CCI-VAE shares all latents between all datasets. VASE use only 8/24 latent dimensions at the end of training. The rest remained as spare capacity for learning on future datasets.

**Learning representations for tasks** We train object identity classifiers (one each for moving Fashion and MNIST) and an object position regressor on top of the latent representation $\mathbf{z} \sim q_\phi(\mathbf{z}|\mathbf{x})$ at regular intervals throughout the continual learning sequence. Good accuracy on these measures would indicate that at the point of measurement, the latent representation **z** contained dataset relevant information, and hence could be useful, e.g. for subsequent policy learning in RL agents. Figure 2 (bottom) shows that both VASE and CCI-VAE learn progressively more informative latent representations when exposed to each dataset $s$, as evidenced by the increasing classification accuracy and decreasing mean squared error (MSE) measures within each stage of training. However, with CCI-VAE, the accuracy and MSE measures degrade sharply once a domain shift occurs. This is not the case for VASE, which retains a relatively stable representation.

**Ablation study** Here we perform a full ablation study to test the importance of the proposed components for unsupervised life-long representation learning: 1) regularisation towards disentangled representations (section 3.2), 2) latent masking (section 3.3 - **A**), 3) environment clustering (section 3.4 - **S**), and 4) "dreaming" feedback loop (section 3.5 - **D**). We use the constraint capacity loss in eq. (3) for the disentangled experiments, and the standard VAE loss [27, 44] for the entangled experiments

| | DISENTANGLED | | | | ENTANGLED | | | |
| | OBJECT ID ACCURACY | | POSITION MSE | | OBJECT ID ACCURACY | | POSITION MSE | |
| ABLATION | MAX (%) | CHANGE (%) | MIN (*1E-4) | CHANGE (*1E-4) | MAX (%) | CHANGE (%) | MIN (*1E-4) | CHANGE (*1E-4) |
|---|---|---|---|---|---|---|---|---|
| - | 88.6 (±0.4) | -15.2 (±2.8) | 3.5 (±0.05) | 24.8 (±13.5) | 91.8 (±0.4) | -12.1 (±0.8) | 4.2 (±0.7) | 10.5 (±2.6) |
| S | 88.9 (±0.5) | -13.9 (±1.9) | 3.4 (±0.05) | 22.5 (±12.2) | 91.7 (±0.4) | -12.2 (±0.03) | 4.5 (±0.8) | 10.9 (±3.1) |
| D | 88.6 (±0.3) | -14.4 (±1.9) | 3.3 (±0.04) | 21.4 (±4.9) | 91.8 (±0.4) | -12.4 (±0.7) | 4.3 (±0.7) | 11.7 (±3.2) |
| A | 86.7 (±1.9) | -24.5 (±1.0) | 3.3 (±0.04) | 67.6 (±107.0) | 88.6 (±0.3) | -19.7 (±0.5) | 4.5 (±0.7) | 47.1 (±26.2) |
| SA | 87.1 (±1.8) | -28.1 (±0.08) | 3.3 (±0.04) | 78.9 (±109.0) | 89.9 (±1.3) | -18.3 (±0.4) | 4.8 (±0.7) | 41.8 (±20.6) |
| DA | 86.3 (±2.5) | -25.2 (±0.5) | 3.3 (±0.04) | 72.2 (±90.0) | 88.8 (±0.3) | -19.4 (±0.4) | 4.6 (±0.7) | 40.2 (±19.2) |
| SD | 88.3 (±0.3) | -12.9 (±1.9) | 3.4 (±0.05) | 20.0 (±3.5) | 91.4 (±0.3) | -11.7 (±0.6) | 4.3 (±0.5) | 11.6 (±1.9) |
| SD-[42] | - | - | - | - | 91.9 (±0.1) | -11.6 (±1.1) | 4.7 (±0.8) | 10.2 (±1.8) |
| **VASE** (SDA) | 88.6 (±0.4) | **-5.4 (±0.3)** | 3.2 (±0.03) | **3.0 (±0.2)** | 91.5 (±0.1) | -6.5 (±0.7) | 4.2 (±0.4) | 3.9 (±1.1) |

Table 1: Average change in classification accuracy/MSE and maximum/minimum average accuracy/MSE when training an object/position classifier/regressor on top of the learnt representation on the moving Fashion → MNIST → moving MNIST sequence. We do a full ablation study of VASE, where D - dreaming feedback loop, S - cluster inference $q(s|\mathbf{x}^s)$, and A - atypicality based latent mask $\mathbf{a}^s$ inference. We compare two versions of our model - one that is encouraged to learn a disentangled representation through the capacity increase regularisation in eq. (3), and an entangled VAE baseline ($\beta = 1$). The unablated disentangled version of VASE (SDA) has the best performance.

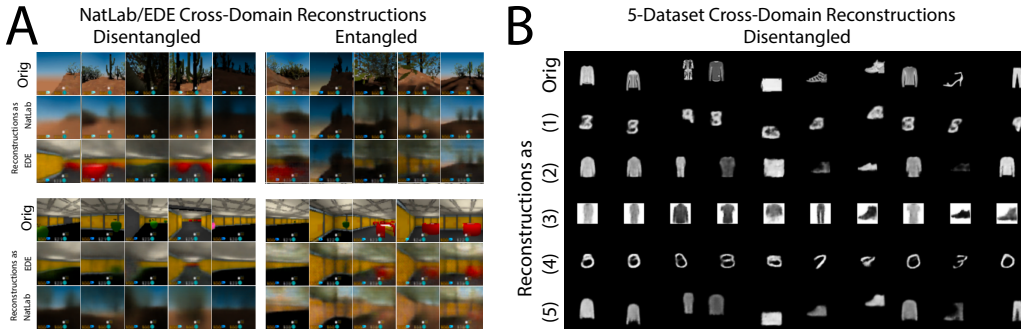

Figure 4: **A** Cross-domain reconstructions on NatLab (outdoors) or EDE (indoors) DM Lab levels. The disentangled VASE finds semantic homologies between the two datasets (e.g. cacti → red objects). The entangled VASE only maps lower level statistics. **B** Cross-domain reconstructions of samples from moving Fashion into each of the five training datasets: moving MNIST (1) → Fashion (2) → inverse Fashion (3) → MNIST (4) → moving Fashion (5).

[21]. For each condition we report the average change in the classification metrics reported above, and the average maximum values achieved (see appendix A.6 for details). Table 1 shows that the unablated VASE (SDA) has the best performance. Note that the entangled baselines perform worse than the disentangled equivalents, and that the capacity constraint of the CCI-VAE framework does not significantly affect the maximal classification accuracy compared to the VAE. It is also worth noting that VASE outperforms the "SD-[42]" condition, which is similar to the only other VAE-based approach to continual learning that we are aware of, see [42]. The difference between the SD and SD-[42] conditions is that the latter also disables the decoder proximity term in eq. (7) to match the model setup in [42]. The only difference between the SD-[42] and the [42] approaches is that we do not use variational inference to learn the value of $s$, opting for a classification-based heuristic instead. This difference is motivated by [42]'s aim to compute a valid variational lower-bound in a life-long setting, while our aim is to learn semantically shared factors. Hence, we sacrifice the probabilistic framework for a better performing heuristic (see Section 3.4 for more details). Our SD-[42] also does not use the Information Gain regularizer of [42], since it would not change the performance of the heuristic.

We have also trained VASE on longer sequences of datasets (moving MNIST → Fashion → inverse Fashion → MNIST → moving Fashion) and found similar levels of performance (see fig. 3).

**Semantic transfer** Here we test whether VASE can learn more sophisticated cross-domain latent homologies than the positional latents on the moving MNIST and Fashion datasets described above. Hence, we trained VASE on a sequence of two visually challenging DMLab-30 [1] [7] datasets: the Exploit Deferred Effects (EDE) environment and a randomized version of the Natural Labyrinth (NatLab) environment (Varying Map Randomized). While being visually very distinct (one being indoors and the other outdoors), the two datasets share many data generative factors that have to do with

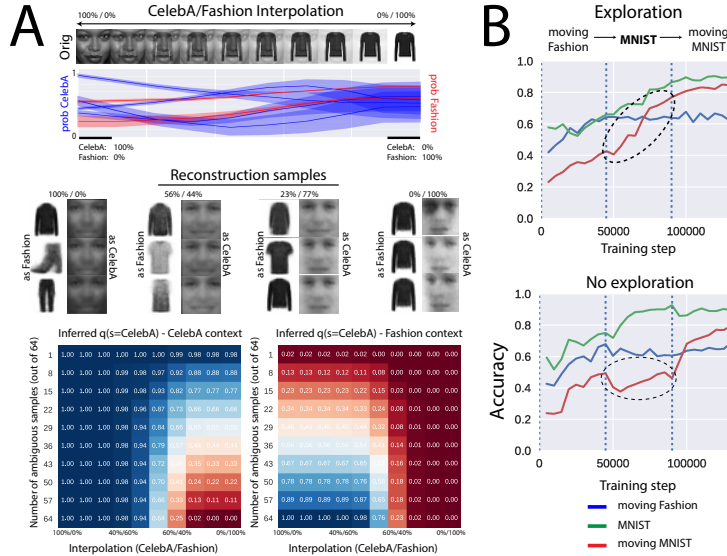

Figure 5: **A** Top: Ambiguous input examples created by using different interpolation weights between samples from CelebA and Fashion, and corresponding inferred parameters $\mu$ (y axis) and $\sigma$ (light colour range) of $q_\phi(\mathbf{z}|\mathbf{x})$; red corresponds to Fashion-specific latents, blue to CelebA-specific latents. Middle: Reconstruction samples $p_\theta(\mathbf{x}^s|\mathbf{z}^s,s)$ for different levels of ambiguity conditioned on either dataset. Bottom: Inferred $q_\psi(s = \text{CelebA}$ given different levels of input ambiguity (x axis) and different number of ambiguous vs real data samples (y axis) for the two datasets. VASE deals well with ambiguity, shows context-dependent categorical perception and uncertainty within its inferred representation parameters. **B** Imagination-based exploration allows VASE to imagine the possibility of moving MNIST digits during static MNIST training by using position latents acquired on moving Fashion. This helps it learn a moving MNIST classifier during static MNIST training without ever seeing real translations of MNIST digits.

the 3D geometry of the world (e.g. horizon, walls/terrain, objects/cacti) and the agent's movements (first person optic flow). Hence, the two domains share many semantically related factors $\mathbf{z}$, but these are rendered into very different visuals $\mathbf{x}$. We compared cross-domain reconstructions of VASE and an equivalent entangled VAE ($\beta = 1$) baseline. The reconstructions were produced by first inferring a latent representation based on a batch from one domain, e.g. $\mathbf{z}^{\text{NatLab}} = q_\phi(\cdot|\mathbf{x}^{\text{NatLab}})$, and then reconstructing them conditioned on the other domain $\mathbf{x}^{\text{xRec}} = q_\theta(\cdot|\mathbf{z}^{\text{NatLab}}, s^{\text{EDE}})$. Figure 4A shows that VASE discovered the latent homologies between the two domains, while the entangled baseline failed to do so. VASE learnt the semantic equivalence between the cacti in NatLab and the red objects in EDE, the brown fog corresponding to the edge of the NatLab world and the walls in EDE (top leftmost reconstruction), and the horizon lines in both domains. The entangled baseline, on the other hand, seemed to rely on the surface-level pixel statistics and hence struggled to produce meaningful cross-domain reconstructions, attempting to match the texture rather than the semantics of the other domain.

Figure 4B demonstrates that VASE also learns to share semantically meaningful factors in the more challenging 5-dataset cross-domain reconstruction task (see Figure 8 and Figure 9 for more details). Note how the position inferred from the moving Fashion dataset is re-used when reconstructing the image as a moving MNIST digit. Furthermore, the clothing type inferred from the moving Fashion is largely shared with the static Fashion and the inverted Fashion datasets. This is not always perfect, however, which highlights one of the limits of our approach. Since the algorithm has only visual information to work with, the "semantics" can sometimes become entangled with the shallow visual statistics – e.g. the clothing categories in the normal and the inverted Fashion are sometimes confused due to the spurious pixel level similarities. The addition of multi-sensory information, or the ability to interact with the environment may help alleviate this problem by integrating different sensory modalities and/or affordances of the environment into the semantic representation.

**Dealing with ambiguity** Natural stimuli are often ambiguous and may be interpreted differently based on contextual clues. Examples of such processes are common, e.g. visual illusions like the Necker cube [38], and may be driven by the functional organisation and the heavy top-down influences within the ventral visual stream of the brain [19, 41]. To evaluate the ability of VASE to deal with

ambiguous inputs based on the context, we train it on a CelebA [33] → inverse Fashion sequence, and test it using ambiguous linear interpolations between samples from the two datasets (fig. 5A, first row). To measure the effects of ambiguity, we varied the interpolation weights between the two datasets. To measure the effects of context, we presented the ambiguous samples in a batch with real samples from one of the training datasets, varying the relative proportions of the two. Figure 5A (bottom) shows the inferred probability of interpreting the ambiguous samples as CelebA $q_\phi(s = \text{celebA}|\mathbf{x})$. VASE shows a sharp boundary between interpreting input samples as Fashion or CelebA despite smooth changes in input ambiguity. Such *categorical perception* is also characteristic of biological intelligence [12, 13, 32]. The decision boundary for categorical perception is affected by the context in which the ambiguous samples are presented. VASE also represents its uncertainty about the ambiguous inputs by increasing the inferred variance of the relevant latent dimensions (fig. 5A, second row).

**Imagination-driven exploration**  If we learn a factor of variation in a past environment (*e.g.*, that objects can move), it may be reasonable to hypothesise that it may also be applicable in the current environment, even if it is not directly observed (*e.g* in an environment with static objects). Given the ability to act on an environment, we may then try to realise an imagined configuration to test whether our hypothesis is correct (*e.g.* try to move the static objects), resulting in a form of imagination-driven exploration. In Appendix A.4 we show how such exploration can be implemented using VASE. Figure 5B shows that on a moving Fashion → MNIST → moving MNIST life-long learning setup, VASE is able to imagine and learn the concept of "moving MNIST digits" before actually experiencing it in the moving MNIST training condition.

# 5   Conclusions

We have introduced VASE, a novel approach to life-long unsupervised representation learning that builds on recent work on disentangled factor learning [21, 9] by introducing several new key components. Unlike other approaches to continual learning, our algorithm does not require us to maintain a replay buffer of past datasets, or to change the loss function after each dataset switch. In fact, it does not require any a priori knowledge of the dataset presentation sequence, since these changes in data distribution are automatically inferred. We have demonstrated that VASE can learn a disentangled representation of a sequence of datasets. It does so without experiencing catastrophic forgetting and by dynamically allocating spare capacity to represent new information. It resolves ambiguity in a manner that is analogous to the categorical perception characteristic of biological intelligence. Most importantly, VASE allows for semantically meaningful sharing of latents between different datasets, which enables it to perform cross-domain inference and imagination-driven exploration. Taken together, these properties make VASE a promising algorithm for learning representations that are conducive to subsequent robust and data-efficient RL policy learning.

# Acknowledgements

We thank Shakir Mohamed and James Kirkpatrick for useful discussions and feedback.

## Footnotes

[1] https://github.com/deepmind/lab/tree/master/game_scripts/levels/contributed/dmlab30#dmlab-30

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
