[Supplementary Material]

# A    Supplemental Details

## A.1    Model details

**Encoder and decoder**    For the encoder we use a simple convolutional network with the following structure: `conv 64` $\to$ `conv 64` $\to$ `conv 128` $\to$ `conv 128` $\to$ `fc 256`, where `conv n_filters` is a $4 \times 4$ convolution with `n_filters` output filters, ReLU activations and stride 2, and similarly `fc n_out` is a fully connected layer with `n_out` units. The output of the fully connected layer is given to a linear layer that outputs the mean $\mu_{\text{enc}}(\mathbf{x})$ and log-variance $\log \sigma_{\text{enc}}^2(\mathbf{x})$ of the encoder posterior $q_\phi(\mathbf{z}|\mathbf{x}) \sim N(\mu_{\text{enc}}(\mathbf{x}), \sigma_{\text{enc}}^2(\mathbf{x}))$. The decoder network receives a sample $\mathbf{z} \sim q_\phi(\mathbf{z}|\mathbf{x})$ from the encoder and outputs the parameters of a distribution $p_\theta(\mathbf{x}|\mathbf{z},s)$ over $\mathbf{x}$. We use the transpose of the encoder network, but we also feed it the environment index $s$ by first encoding it with a one-hot encoding (of size `max_environments`, which is a hyperparameter), and then concatenating it to $\mathbf{z}$. For most of the experiments, we use a product of independent Bernoulli distributions (parametrised by the mean) for the decoder distribution $p_\theta(\mathbf{x}|\mathbf{z},s)$. In the DM Lab experiments we use instead a product of Gaussian distributions with fixed variance. We train the model using Adam [25] with a fixed learning rate 6e-4 and batch size 64.

**Environment inference network**    We attach an additional fully connected layer to the last layer of the encoder (gradients to the encoder are stopped). Given an input image $\mathbf{x}$, the layer outputs a softmax distribution $q_\psi(s|\mathbf{x})$ over `max_environments` indices, which tries to infer the most likely index $s$ of the environment from which the data is coming, assuming the environment was already seen in the past. Notice that we always know the (putative) index of the current data ($\hat{s}$ in Equation (6), also see Appendix A.2), so that we can always train this layer to associate the current data to the current index. However, to avoid catastrophic forgetting we also need to train on hallucinated data from past environments. Assuming $\hat{s}$ is the current environment and $m$ is the total number of environments seen until now, the resulting loss function is given by:

$$\mathcal{L}_{\text{env}} = \underbrace{\mathbb{E}_\mathbf{x}[-\log(q_\psi(\hat{s}|\mathbf{x}))]}_{\text{Classification loss on current data}} + \underbrace{\mathbb{E}_{\hat{s} \neq s < m} \mathbb{E}_{\mathbf{x}' \sim p_{\theta'}(\mathbf{x}'|\mathbf{z}',s)}[-\log q_\psi(s|\mathbf{x}')]}_{\text{Classification loss on hallucinated data}},$$

where the hallucinated data $\mathbf{x}'$ in the second part of the equation is generated according to Section 3.5, and the expectation over $s$ is similarly done through Monte Carlo sampling.

## A.2    Extra algorithm implementation details

**Atypical latent components**    The atypicality $\alpha_n$ of the component $z_n$ on a batch of samples $\mathbf{x}_1, \ldots, \mathbf{x}_B$ is computed using a KL divergence from the marginal posterior over the batch to the prior according to Equation (4). In practice it is not convenient to compute this KL divergence directly. Rather, we observe that the marginal distribution of the latent samples $\frac{1}{B} \sum_{b=1}^{B} q_\phi(z_n|\mathbf{x}_b)$ is approximately Gaussian. We exploit this by fitting a Gaussian to the latent samples $\mathbf{z}$ and then computing in closed-form the KL-divergence between this approximation of the marginal and the unit Gaussian prior $p(\mathbf{z}) = \mathcal{N}(0,1)$.

Recall from Section 3.3 that we deem a latent component $z_n$ to be active ($a_n = 1$) whenever it is typical, that is, if $\alpha_n < \lambda$. However, since the atypicality is computed on a relatively small batch of $B$ samples, $\alpha_i$ may be a noisy estimate of atypicality. Hence we introduce the following filtering: we set $\alpha_n = 1$ if $\alpha_n > \lambda_1$ and $\alpha_n = 0$ if $\alpha_n < \lambda_0$, with $\lambda_0 < \lambda_1$. If $\lambda_0 < \alpha_n < \lambda_1$, we leave $\alpha_n$ unchanged.

**Used latent components**    We say that a factor $z_n$ is not used by the environment $s$ if the reconstruction $p_\theta(\mathbf{x}|\mathbf{z},s)$ does not depend on $z_n$. To measure this, we find the maximum amount of noise we can add to $z_n$ without changing the reconstruction performance of the network. That is, we optimise

$$\Sigma = \underset{\Sigma = \text{diag}(\sigma_1, \ldots, \sigma_N)}{\text{argmin}} \quad \mathbb{E}_{\epsilon \sim \mathcal{N}(0,\sigma)}[-\log p_\theta(\mathbf{x}|\mathbf{z}^\epsilon, s)] - \log|\Sigma|$$

where $\mathbf{z}_n^\epsilon = (1 - \delta_{nm})z_n + \delta_{nm}(z_n + \epsilon)$. If $\sigma_n' > T$ for some threshold $T$, we say that $z_n$ is unused. We generally observe that components are either completely unused $\sigma_n = 0$, or else $\sigma_n$ is very large. Therefore, picking a threshold is very easy and the precise value does not matter. We only compare the atypicality masks in eq. (6) for the used latents.

**Environment index**    Expanding on the explanation in Section 3.4, let $L_s(\mathbf{x}) = \mathbb{E}_{\mathbf{z} \sim q_\phi(\mathbf{z}|\mathbf{x})}[-\log p_\theta(\mathbf{x}|\mathbf{z}^s, s)]$ be the reconstruction loss on a batch $\mathbf{x}$ of data, assuming it comes from the environment $s$. Let $\tilde{L}_s$ be the average reconstruction loss observed in the past for data from environment $s$. Let $m$ be the number of datasets observed until now. Let $\mathbf{u}^s$ be a binary vector of used units computed with the method described before.

We run the auxiliary environment inference network (Appendix A.1) on each sample from the batch $\mathbf{x}$ and take the average of all results in order to obtain a probability distribution $q(s|\mathbf{x})$ over the possible environment $s$ of the batch $\mathbf{x}$, assuming it has already been seen in the past. Let $\hat{s} = \text{argmax}_s q(s|\mathbf{x})$ be the most likely environment, which is our candidate for the new environment. If the reconstruction loss $L_{\hat{s}}(\mathbf{x})$ (assuming $s = \hat{s}$) is significantly larger (see

| | **DISENTANGLED** | | | | | | **ENTANGLED** | | |
|---|---|---|---|---|---|---|---|---|---|
| EXPERIMENT | $\gamma$ | $C_{\text{MAX}}$ | $\delta C$ | $\lambda$ | $\kappa$ | $\tau$ | $\lambda$ | $\kappa$ | $\tau$ |
| ABLATION STUDY (150K) | 100.0 | 35.0 | 6.3e-6 | 0.6 | 1.5 | 500 | | | |
| FIVE DATASETS (150K) | 100.0 | 35.0 | 6.3e-6 | 0.6 | 1.5 | 5000 | | | |
| CELEBA → INVERTED FASHION (30K) | 200.0 | 20.0 | 1.7e-5 | 0.8 | 1.1 | 500 | | | |
| NATLAB → EDE (60K) | 200.0 | 25.0 | 1e-5 | 2.0 | 1.1 | 5000 | 20.0 | 1.1 | 5000 |
| IMAGINATION-DRIVEN EXPLORATION (45K) | 200.0 | 35.0 | 0.7e-5 | 0.7 | 1.5 | 500 | | | |

Table 2: Hyperparameter values used for the experiments reported in this paper. Values in brackets after the experiment name indicate the number of training steps used per dataset.

Algorithm 1) than the average loss for the environment $\hat{s}$, we decide that the data is unlikely to come from this environment, and hence we allocate a new one. If the reconstruction is good, but some of the used components (given by $\mathbf{u}$) are atypical, we still allocate a new environment. Otherwise, we assume that the data indeed comes from $\hat{s}$.

---

**Algorithm 1** Infer the environment index $s$ from a batch of data

---

$\hat{s} \leftarrow \text{argmax}_s \mathbb{E}_{\mathbf{z} \sim q_\phi(\mathbf{z}|\mathbf{x})}[-\log p_\theta(\mathbf{x}|\mathbf{z}^s, s)]$

**if** $L_{\hat{s}} > \kappa \tilde{L}_{\hat{s}}$ **then**
    $s \leftarrow m+1$
**else if** $a^{\hat{s}} \odot \mathbf{u}^s \neq \mathbf{a}(\mathbf{x}) \odot \mathbf{u}^{\hat{s}}$ **then**
    $s \leftarrow m+1$
**else**
    $s \leftarrow \hat{s}$

---

## A.3 Hyperparameter sensitivity

Table 2 lists the values of the hyperparameters used in the different experiments reported in this paper.

For all experiments we use `max_environments = 7`, and we increase $C$ in eq. (3) linearly by $\delta C \cdot C_{\max}$ per step (starting from 0) until it reaches $C_{\max}$, at which point we keep $C$ fixed at that value. In the loss function eq. (8), the dreaming loss was re-weighted, with the full loss being:

$$\mathcal{L}(\phi, \theta) = \underbrace{\mathbb{E}_{\mathbf{z}^s \sim q_\phi(\cdot|\mathbf{x}^s))}[-\log p_\theta(\mathbf{x}|\mathbf{z}^s, s)] + \gamma|\mathbb{KL}(q_\phi(\mathbf{z}^s|\mathbf{x}^s)||p(\mathbf{z})) - C|^2}_{\text{MDL on current data}} +$$

$$+ \underbrace{\mathbb{E}_{\mathbf{z}, s', \mathbf{x}'}\Big[ \alpha D[q_\phi(\mathbf{z}|\mathbf{x}'), q_{\phi'}(\mathbf{z}'|\mathbf{x}')] + \beta D[q_\theta(\mathbf{x}|\mathbf{z}, s'), q_{\theta'}(\mathbf{x}'|\mathbf{z}, s')] \Big]}_{\text{``Dreaming'' feedback on past data}}. \tag{9}$$

The values $\alpha = 1000$ and $\beta = 20$ were used for all experiments, except for in the hyperparameter sweep.

For the ablation study we ran a hyperparameter search using the full model, and used the best hyperparameters found for all experiments. We list the search ranges and our observed sensitivity to these hyperparameters next:

- $\gamma$ = coefficient for the capacity constraint – $\{50, 100, 200\}$ – found not to be very sensitive.
- $C_{\max}$ = final value of $C$ – $\{20, 35, 50\}$ – classification accuracy increased significantly for capacity from 20 to 35.
- $\lambda$ = atypicality threshold – $\{0.4, 0.6, 1, 2\}$ – lower threshold led to more latent freezing. Classification performance was not very sensitive to this.
- $\tau$ = update frequency reference network – $\{500, 1000, 2000, 5000\}$ – found not to be very sensitive.
- $\alpha$ = weight for encoder loss in "dreaming" loop – $\{10, 20, 40\}$ – found not to be very sensitive.
- $\beta$ = weight for decoder loss in "dreaming" loop – $\{500, 1000, 2000\}$ – found not to be very sensitive.

## A.4 Imagination-based exploration experiments

Once we learn the concept of moving objects in one environment, it is reasonable to imagine that a novel object encountered in a different environment can also be moved. Given the ability to act, we may try to move the object to realise our hypothesis. We can use such imagination-driven exploration to augment our experiences in an environment to learn a richer representation. Notice however, that such imagination requires a compositional representation that allows for novel yet sensible recombinations of previously learnt semantic factors. We now

investigate whether VASE can use such imagination-driven exploration to learn better representations using a sequence of three datasets: moving Fashion $\rightarrow$ MNIST $\rightarrow$ moving MNIST.

We model a very simple interaction with the environment where the agent can translate the observed object (in our case the input image). The agent is trained as follows: a random $\mathbf{z}^*$ is sampled from the prior $p(\mathbf{z})$. Given an observation $\mathbf{x}$ from the environment, the agent needs to pick an action $g(\mathbf{z}^*, \mathbf{x})$ (in our case a translation) in such a way that the encoding $\mathbf{z} \sim q_\phi(\mathbf{z}|g \cdot \mathbf{x})$ of the new image $g \cdot \mathbf{x}$ is as close as possible to $\mathbf{z}^*$. That is, we minimise the loss

$$\mathcal{L}_{\text{agent}} = \mathbb{E}_{\mathbf{x} \sim p(\mathbf{x})} \mathbb{E}_{\mathbf{z}^* \sim p(\mathbf{z})} \mathbb{E}_{\mathbf{z} \sim q(\mathbf{z}|g(\mathbf{z}^*, \mathbf{x}) \cdot \mathbf{x})} \|\mathbf{z}^* - \mathbf{z}\|^2.$$

The agent can then be used to "explore" the current environment. Given an image $\mathbf{x}$ from the current environment, and a random configuration $\mathbf{z}^*$ sampled from the prior $p(\mathbf{z})$, and we let the agent act on the environment in order to realise the imagined configuration. The result of this action is a new image $\mathbf{x}^* = g(\mathbf{z}, \mathbf{x}) \cdot \mathbf{x}$. The image $\mathbf{x}^*$ is then added to the training data, so that the encoder will learn from a more diversified set of inputs than the images $\mathbf{x} \sim p(\mathbf{x})$ observed passively.

We apply this method to the sequence moving Fashion $\rightarrow$ MNIST $\rightarrow$ moving MNIST. During the first stage (moving Fashion), VASE learns the concepts of position and the shape of Fashion sprites. It also learns how to move the sprites to reach a hallucinated state $z^*$, thus acquiring the ability to manipulate the environment in order to improve exploration. Later, when presented with fixed MNIST digits, VASE implements an imagination-based augmentation of the input data by moving MNIST digits to different positions using the learnt policy. In other words, VASE can imagine the existence of moving MNIST, and act on the environment to realise this configuration, before actually experiencing the moving MNIST dataset. Indeed, fig. 5C shows that when we train a moving MNIST classifier during the static MNIST training stage, the classifier is able to achieve good accuracy in the *imagination-driven exploration* condition, highlighting the benefits of imagination-driven data augmentation.

**Details of the policy network** The policy network first processes the input image $\mathbf{x}$ (of size $64 \times 64$) though four convolutional $4 \times 4$ layers with 16 filters, stride 2 and ReLU activations. The resulting vector is concatenated with the target $\mathbf{z}^*$, and fed to a 1-hidden layer fully connected network that outputs the parameters of the 2D translation $g(\mathbf{x}, \mathbf{z}^*)$ which needs to be applied to the image. We use a `tanh` output to ensure the translation is always in a sensible range. Once these parameters are obtained, the transformation is applied to the image $x$ using a Spatial Transformer Network (STN) [23], thus obtaining a translated image $\mathbf{x}^* = g(\mathbf{z}^*, \mathbf{x}) \cdot \mathbf{x}$. We can now finally compute the resulting representation $\mathbf{z} \sim q(\mathbf{z}|g(\mathbf{z}^*, \mathbf{x}) \cdot \mathbf{x})$. We compare this with the target configuration $\mathbf{z}^*$ to obtain the loss $\mathcal{L}_{\text{agent}}$ for the policy, which tries to minimise the distance between $\mathbf{z}$ and the target $\mathbf{z}^*$. Notice that the whole operation is fully differentiable, thanks to the properties of the STN, so the policy loss can be minimised by stochastic gradient descent. In our experiments, the policy is trained in parallel with the main model.

## A.5 Dataset processing

**DM Lab** We used an IMPALA agent trained on all DM-30 tasks [11] to generate data. We take observations of this optimal agent (collecting rewards according to the task descriptions explained in https://github.com/deepmind/lab/tree/master/game_scripts/levels/contributed/dmlab30), on randomly generated episodes of Exploit Deferred Effects and NatLab Varying Map Randomized; storing them as $111 \times 84 \times 3$ RGB tensors. We crop the right-most 27 pixels out to obtain a $84 \times 84$ image (this retains the most useful portion of the original view), which are finally scaled down to $64 \times 64$ (using `tf.image.resize_area`).

**CelebA $\rightarrow$ Inverse Fashion** To make CelebA compatible with Fashion, we convert the CelebA images to grayscale and extract a patch of size $32 \times 32$ centered on the face. We also invert the colours of Fashion so that the images are black on a white background, and slightly reducing the contrast, in order to make the two datasets more similar, and hence easier to confuse after mixing.

## A.6 Quantifying catastrophic forgetting

We train on top of the representation $\mathbf{z} \sim q_\phi(\mathbf{z}|\mathbf{x})$ a simple 2-hidden layers fully connected classifier with 256 hidden units per layer and ReLU activations. At each step while training the representation, we also train a separate classifier on the representation for each environment, using Adam with learning rate 6e-4 and batch size 64. This classifier training step does not update the weights in the main network.

For each ablation type we reported the average classification accuracy (or regression MSE) score obtained by 20 replicas of the model, all with the best set of hyperparameters discovered for the full model. We quantified catastrophic forgetting by reporting the average difference between the maximum accuracy obtained while VASE was training on a particular dataset and the minimum accuracy obtained for the dataset afterwards.

## A.7 Additional results

We present additional experimental results and extra plots for the experiments reported in the main paper here. Fig. 6 and table 3 show latent traversals and quantitative evaluation results for an ablation study on VASE trained

|  | **DISENTANGLED** | | **ENTANGLED** | |
| CONFIGURATION | AVG. DECREASE (%) | AVG. MAX (%) | AVG. DECREASE (%) | AVG. MAX (%) |
|---|---|---|---|---|
| DA | -7.9 | 90.5 | -12.1 | 90.9 |
| SD | -2.2 | 91.0 | -4.3 | **92.1** |
| S | -3.9 | 90.8 | -8.4 | 91.5 |
| A | -9.6 | 90.2 | -10.1 | 91.7 |
| SA | -5.9 | 90.0 | -10.3 | 91.0 |
| - | -4.4 | 91.1 | -6.6 | 92.6 |
| D | -6.0 | 90.5 | -6.9 | 91.4 |
| **VASE** (SDA) | **-0.9** | 90.3 | -2.5 | 91.2 |

Table 3: Average drop in classification accuracy and maximum average accuracy when training an object classifier on top of the learnt representation on the MNIST → Fashion → MNIST sequence. We do a full ablation study of VASE, where D - dreaming feedback loop, S - cluster inference $q(s|\mathbf{x}^s)$, and A - atypicality based latent mask $\mathbf{a}^s$ inference. We compare two versions of our model - one that is encouraged to learn a disentangled representation through the capacity increase regularisation in eq. (3), and an entangled VAE baseline ($\beta = 1$). The unablated disentangled version of VASE (SDA) has the best performance.

Figure 6: Latent traversals for VASE trained on MNIST → Fashion → MNIST, and DM Lab levels NatLab → EDE.

on the MNIST → Fashion → MNIST sequence. Fig. 6 also shows traversals for VASE trained on the DM Lab levels NatLab → EDE. This is the model reported in section 4. Fig. 7 shows cross-dataset reconstructions for VASE trained on the moving Fashion → MNIST → moving MNIST sequence described in the ablation study in section 4. Figs. 8, 9 shows latent traversals and cross-dataset reconstructions for VASE trained on the moving MNIST → Fashion → inverted Fashion → MNIST → moving Fashion sequence described in the main text.

Moving Fashion/Moving MNIST
Cross-Domain Reconstructions

Figure 7: Cross-domain reconstructions for the entangled and disentangled versions of VASE (as described in section 4) trained on moving Fashion → MNIST → moving MNIST. We see that the entangled baseline forgets moving Fashion by the end of training.

# Latent Traversals

Figure 8: Latent traversals for VASE trained on a sequence of moving MNIST → Fashion → inverse Fashion → MNIST → moving Fashion.

# Cross-Domain Reconstructions

Figure 9: Cross-domain reconstructions for the VASE trained on moving MNIST → Fashion → inverted Fashion → MNIST → moving Fashion.