[Reviews · NeurIPS 2018]

Reviewer 1



This paper describes a new variational autoencoder with a mechanism for detecting continuous changes in latent environments, which, combined with a "representational capacity" term in its loss function, allows it to learn environment-specific sets of relevant generative factors. The model seems reasonable, and while I'm aware of VAEs that appear to learn cleaner and more easily-interpreted disentangled representations, they're susceptible to catastrophic forgetting. The evaluation supports the key claim that the model is more resistant to forgetting than the baseline, and demonstrates some interesting properties. The topic is interesting, the writing is generally clear, the key claims seem to be justified, and I didn't see any serious weaknesses in the evaluation. That said, I'm not as familiar with modern VAEs as other reviewers are likely to be. The section on "imagination-driven exploration" could have been clearer, but I believe the paper is likely to interest a reasonably large audience at NIPS even if we set it aside.

Reviewer 2



The authors present an algorithm for lifelong representation learning that adapts variational autoencoders to the lifelong learning setting. The framework is presented as a full generative process, where a set of latent factors are (selectively) shared across tasks, and the tasks themselves are generated by an unknown distribution. The algorithm optimizes for the reconstruction error with a regularization based on the MDL principle that has been studied for learning disentangled representations. The algorithm automatically detects distribution shifts (i.e., task changes) and avoids catastrophic forgetting by "hallucinating" data for previous tasks while training on a new one. The authors show empirically that their algorithm is able to extract relevant semantic knowledge from one task and transfer it to the next. Overall, I think this is a very good paper. The topic of lifelong (deep) representation learning addressed in the paper is very underdeveloped (which the authors rightfully point out), which I believe makes this a valuable contribution. My main concern with this paper is the lack of baseline comparisons--[40] is mentioned as a related approach, and there are also multiple other non-deep representation learning approaches (e.g., online dictionary learning) and non-representation learning approaches to lifelong learning that could address some of the problems in the experiments section. The algorithm selectively shares latent components via a boolean mask, which is inferred by keeping a threshold on how much the posterior of the component given the data diverges from the prior on the component. There seems to be an important connection between this masking and sparse coding, a popular mechanism in lifelong learning, and so I believe the authors should discuss this relationship and provide a qualitative comparison. The algorithm infers the task/environment by comparing the reconstruction error of the most likely environment on the new data and on its own data, and by comparing the boolean masks. The authors justify why merely comparing the errors is not enough, but not why merely comparing the masks is not. The latter seems like a viable option, so it seems that the authors should either attempt using it or explain why they don't. Finally, the algorithm avoids catastrophic forgetting by generating samples from the partially learned generative process and imposing a penalty for changing the encoder and decoder too much for previous datasets. This approach, instead of using actual data from the previous tasks, is substantially more data efficient, which is key in a lifelong learning context. One concern I have is that for this penalty, the authors use different distribution distances for two distributions, but do not justify this choice. The experimental setting is very good, first providing general performance measures and then adding other experiments that are useful for seeing the effect of each of the separate parts of the framework. The ablation study addresses one big potential concern when presenting an algorithm with so many components, which was good. I also thought the semantic transfer experiment was helpful to show the potentila of the algorithm for transfering knowledge across highly varied tasks. However, the authors compare only to a single-task learning baseline in the main results. They point out [40] as the only other lifelong variational autoencoder, but limit their comparison to the SD condition in Table 1 claiming that it is similar to [40], without providing any explanation about how it is similar. In fact, there is no description or qualitative comparison to [40] in the paper. Since there is apparently no direct comparison, the author's claim that their method outperforms the only related approach seems invalid. The readability of the paper could be improved by providing an intuitive explanation of the terms in the equations before introducing the equations themselves. Typos - Line 126: "... ignore informative z_n irrelevant..." - Equation 4: x_b^s is not introduced --> what does subscript b stand for?

Reviewer 3



By defining a generative process of images based on latent environments and shared latent generative factors which will be chosen depending on the environment, the authors propose an algorithm for life-long unsupervised representation learning. Reasonable inference procedures for the essential elements (the generative factors, the latent mask, and the environment) are proposed, and the training is done with a loss inspired in the Minimum Description Length principle and a "dreaming" feedback loop similar to the one in Shin et al. (2017). The algorithm is a mix of several components for which the authors well state the motivations for them to be there. The work is mostly one of modeling, and no theoretical evidence supports the claims. All the evidence is experimental, which is very dense as a consequence of the several components and claims. Even though this work is of high quality, the quality comes from the engineering part. The authors proposed to tackle several scientific questions at once by mixing several components. In the end, the algorithm works and improves the state of the art, but there is not a clear principle that has been studied. However, this algorithm might be a good starting point to focus on one question such as to which extent the algorithm exhibits "semantically" meaningful sharing of latents between different datasets? Of course, one part of the experimental section is devoted to showing evidence for this question, but since there are several other claims, the evidence is superficial, there is no an attempt to falsify the claims to see if they are robust or at least to show the weaknesses of the algorithm. Doing this for each one of the claims would probably result in a paper with many more pages; this is why it would be desirable to focus on one specific question and answer it thoroughly. Studying better one aspect would increase the scientific quality of the paper, and improve the significance of the article. The paper is written clearly, except for a few phrases which are not well explained I believe. For instance, in line 169, what motivates the use of the Wasserstein distance for the encoder and the KL divergence for the decoder? And the other examples are mostly about interpretation. To show clearly the evidence and the claims supported it might be a better idea to use words whose meaning in the given context is not up to ambiguity. For example, in line 268, "VASE can imagine the existence of moving MNIST before actually experiencing it" this phrase corresponds to an interpretation of the fact that when training a classifier for moving MNIST during the static MNIST training stage, the classifier can achieve an accuracy that is not random. The reasons that explain this fact might not have anything to do with "imagination" in the sense of artificially generated; maybe they could come from a translation invariant property of the classifier. The translation invariant explanation might not be correct, it is just an example to make a point of avoiding interpretations that implicitly make claims not supported by evidence, or at least for which the evidence has not been clearly explained. To my knowledge, the idea of considering latent variables that might be or might not be shared across different environments and the idea of considering latent environments are new and original. I believe there might be exciting potential in exploring the latent environments idea profoundly. In my view, it would be more significant to focus on this part of the paper, giving clear empirical evidence for the proposed claims without filling the experimental part densely trying to do everything at once. Update: I still think that it is better to tackle one specific question at a time; however, I find the number of experiments and the different ideas valuable, even if it is not an in-depth study (in my opinion) but an illustration of the potential of the algorithm. Also, as explained by the authors in the rebuttal, I understand that the several experiments are heavily linked (as a consequence of the nature of the algorithm, which is a mix of several components) which implies that it is not straightforward to isolate one specific question. I hope that the rearrangements in the experimental section that the authors mention in their rebuttal will help to illustrate the studied points better. For all these reasons, I have changed my score from 4 to 6.